# [Re] $p$-Poisson surface reconstruction in curl-free flow from point clouds

## Abstract

This study presents a reproducibility analysis of the $p$-Poisson surface reconstruction method presented by Park et al. (NeurIPS 2023). The method utilizes the $p$-Poisson equation and a curl-free constraint for improved surface reconstruction from point clouds, claiming significant advancements over existing implicit neural representation techniques. This study evaluates the reproducibility and generalizability of the results reported in the original paper, focusing on the evaluation using the Surface Reconstruction Benchmark (SRB) dataset. The neural network architecture and training procedures are entirely re-implemented from scratch, emphasizing correctness and efficient execution. While the replication generally outperforms the four alternative methods mentioned in the original paper, the distance results reported in the original paper fail to be reproduced by the re-implementation. Notably, training with the code published in the original paper yields similar results to the reproduced results, still deviating from the findings presented in the original paper. The presented implementation demonstrates a significant improvement in training performance, achieving a five-fold acceleration in training times compared to the code used in the original paper by vectorizing the gradient calculations and leveraging just-in-time compilation of the training loop, which gives an actionable insight for others to explore and integrate such optimizations into their machine learning code. The re-implementation is available at [1].

## 1  Introduction

Extensive research in computer vision and graphics has focused on the task of surface reconstruction from unorganized point clouds. As traditional mesh-based methods lack flexibility and do not ensure watertight surfaces, implicit function-based approaches such as signed distance functions (SDFs) or occupancy functions offer a solution to this problem. The rise of deep learning methods has led to the introduction of implicit neural representations (INRs), utilizing neural networks to parameterize implicit functions for expressive reconstructions. Early INRs formulated the problem as supervised regression and faced challenges with ground-truth distance values. Some methods employ partial differential equations (PDEs) like the eikonal equation to alleviate the need for 3D supervision, but they struggle with non-unique solutions and reliance on accurate normal vectors, which may be unavailable in raw point cloud data. Additionally, these methods are sensitive to noise and outliers, limiting their effectiveness in reconstructing fine details or realistic surfaces without normal vectors. This study aims to evaluate the reproducibility of the method termed $p$-**P**oisson equation based **I**mplicit **N**eural representation with **C**url-free constraint (PINC) introduced by Park et al. (2023) which purportedly achieves significant improvements over comparable INRs without relying on supplementary information such as surface normals.

## 2  Scope of reproducibility

The goal of this study is to evaluate the reproducibility of the results presented by Park et al. (2023) and the performance of the proposed method for surface reconstruction from unorganized point clouds, focusing on

---

[1]https://anonymous.4open.science/r/pinc-B7CD

the efficient implementation of the PDE-based variable-splitting strategy and training process. Furthermore the proposed approach's surface reconstruction performance on the surface reconstruction benchmark (SRB) dataset (Berger et al., 2013) is analyzed. The original paper claims that the proposed method outperforms existing implicit neural representation approaches in terms of the metrics Chamfer and Hausdorff distances, robustness to noise, and handling of incomplete observations. This reproducibility study also offers a novel implementation of the proposed method using JAX which significantly improves training speed by vectorizing the multiple gradient calculations of the model and by writing the entire training loop using primitives that can be transformed into low-level instructions using the XLA compiler.

## 3 Methodology

The original paper by Park et al. (2023) introduces a novel approach to surface reconstruction from point clouds using INRs. The proposed method leverages the $p$-Poisson equation and a curl-free constraint to enhance the accuracy and robustness of the reconstructed surfaces. Unlike comparable methods that require additional information such as surface normals, the proposed approach learns the SDF implicitly, allowing more flexible and accurate surface reconstructions.

### 3.1 Problem Formulation

Given an unorganized point cloud $\mathcal{X} = \{\boldsymbol{x}_i\}_{i=1}^N$ sampled from a closed surface $\Gamma$, the goal is to learn a signed distance function $u : \mathbb{R}^3 \to \mathbb{R}$, where the zero level set of $u$ accurately represents the surface $\Gamma$, such that $\Gamma = \{\boldsymbol{x} \in \mathbb{R}^3 \mid u(\boldsymbol{x}) = 0\}$. In the following equations, $\Omega$ represents a subset of the entire definition space of $u(\boldsymbol{x})$ for which we sample points to constrain the optimization.

$p$-**Poisson Equation** The $p$-Poisson equation serves as the foundation for PINC, allowing us to model the signed distance function with high precision. The decision to utilize the $p$-Poisson equation in PINC stems from its ability to provide a unique and stable solution to the surface reconstruction problem. Unlike methods that rely solely on the eikonal equation, which can lead to non-unique solutions and numerical instability, the $p$-Poisson equation offers a more robust framework for approximating the signed distance function. By formulating the surface reconstruction problem as a minimization of the $p$-Poisson equation, PINC can achieve more accurate and reliable reconstructions of surfaces from point clouds. The equation is defined as:

$$\min_u \int_\Gamma |u| d\boldsymbol{x} + \lambda_1 \int_\Omega \left| \nabla_{\boldsymbol{x}} \cdot \left( \|\nabla_{\boldsymbol{x}} u\|^{p-2} \nabla_{\boldsymbol{x}} u \right) + 1 \right| d\boldsymbol{x}, \tag{1}$$

where $\lambda_1 > 0$ is a weighting hyperparameter, $\Gamma$ represents the surface, and $\Omega$ denotes a subset of the domain where points are sampled for optimization. This equation seeks to minimize the integral of the absolute value of the SDF over the surface $\Gamma$, subject to a regularization term in the domain $\Omega$. In the PINC framework, this equation is redefined into a loss function of the form:

$$\mathcal{L}_{p\text{-Poisson}} = \int_\Gamma |u| d\boldsymbol{x} + \lambda_1 \int_\Omega \|\nabla_{\boldsymbol{x}} u - G\|^2 d\boldsymbol{x} \tag{2}$$

where $G$ is introduced using variable splitting, decomposing a complex minimization problem into simpler sub-problems and coupling them. $G$ is derived from the auxiliary neural network output vector $\Psi$, along with a fixed function $F(\boldsymbol{x}) = \frac{\boldsymbol{x}}{3}$, chosen such that $\nabla_{\boldsymbol{x}} \cdot F = 1$. This expression of $G$ acts as a hard constraint for $G$ and is formulated as

$$G = \frac{\nabla_{\boldsymbol{x}} \times \Psi - F}{\|\nabla_{\boldsymbol{x}} \times \Psi - F\|^{\frac{p-2}{p-1}}}. \tag{3}$$

This auxiliary variable $G$ plays a crucial role within the PINC framework since it serves as a proxy for $\nabla_{\boldsymbol{x}} u$, which makes it possible to incorporate the hard contract set on $G$ into the optimization. This could not have been achieved solely through the direct automatic differentiation of $\nabla_{\boldsymbol{x}} u$ without the use of variable splitting.

**Curl-Free Constraint** In the original paper, the authors argue that to enhance the accuracy of surface reconstructions from point clouds, a curl-free constraint should be applied to the auxiliary variable $G$, which

represents the gradient of the SDF, to ensure that $G$ forms a conservative vector field. A conservative vector field condition, expressed as $G = \nabla_{\boldsymbol{x}} u$, indicates that $G$ is curl-free ($\nabla_{\boldsymbol{x}} \times G = 0$) if it can be written as the gradient of some scalar potential function $u$.

Implementing a direct penalty for the curl of $G$ to enforce this constraint ($\int_{\Omega} \|\nabla_{\boldsymbol{x}} \times G\|^2 dx$) is claimed to introduce computational challenges and a complex loss landscape due to high-order derivatives required by automatic differentiation (Park et al., 2023). To mitigate these issues, an additional auxiliary variable $\tilde{G}$ is introduced to satisfy both $G = \tilde{G}$ and the curl-free condition $\nabla_{\boldsymbol{x}} \times \tilde{G} = 0$, through the loss function

$$\mathcal{L}_{\text{PINC}} = \mathcal{L}_{p\text{-Poisson}} + \lambda_2 \int_{\Omega} \left\| G - \tilde{G} \right\|^2 dx + \lambda_3 \int_{\Omega} \left\| \nabla_{\boldsymbol{x}} \times \tilde{G} \right\|^2 dx. \tag{4}$$

where $\lambda_2, \lambda_3 > 0$ are weighting hyperparameters.

The optimality conditions suggest that $\tilde{G}$ should have a unit norm, as dictated by the Eikonal equation. To simplify adherence to this non-convex equality constraint, it is relaxed to a convex condition $\|\tilde{G}\| \leq 1$, using a projection that maps the auxiliary output $\tilde{\Psi}$ to $\tilde{G}$ within the three-dimensional unit sphere:

$$\tilde{G} = \frac{\tilde{\Psi}}{\max\left\{1, \|\tilde{\Psi}\|\right\}}, \tag{5}$$

How the neural network structure relates to the auxiliary variable is summarized in figure 1.

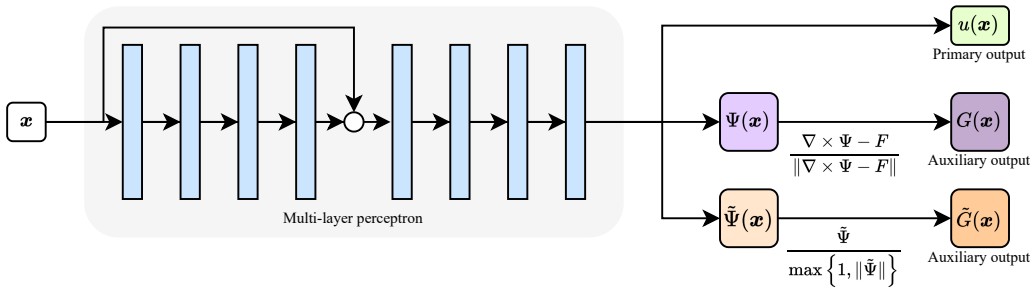

Figure 1: The visualization of the augmented network structure with two auxiliary variables.

**Variable-Splitting Strategy**

PINC adopts a variable-splitting strategy to simplify the optimization process; splitting the network into multiple outputs, as illustrated in Fig. 1, for the signed distance function and its gradient. In the original paper, it is argued that this approach enables more effective training and better adherence to the $p$-Poisson and curl-free constraints.

### 3.2 Loss Function

Real point clouds from range scanners often have incomplete data due to occlusions and concavities, resulting in holes. Estimating accurate closed surfaces becomes challenging, requiring a strategy to interpolate across gaps and reconstruct the surface cohesively.

PINC's approach is to minimize the surface area of the zero-level set, which is encapsulated in the final augmented loss function:

$$\mathcal{L}_{\text{total}} = \mathcal{L}_{\text{PINC}} + \lambda_4 \int_{\Omega} \delta_{\epsilon}(u) \left\| \nabla_{\boldsymbol{x}} u \right\| dx \tag{6}$$

where $\lambda_4 > 0$ is the weighting hyperparameter of the surface area minimization component in the total loss function, and $\delta_{\epsilon}(x) = 1 - \tanh^2\left(\frac{x}{\epsilon}\right)$ represents a smoothed Dirac delta function with a smoothing parameter

$\epsilon > 0$. This addition aims to guide the surface reconstruction process by encouraging the minimization of the zero-level set area of $u$, thereby promoting a more coherent filling of the missing parts of the scanned point cloud.

To implement the loss function in code, all integrals are approximated using Monte Carlo integration. This involves simply replacing the integrals with sums over the correct data points.

### 3.3 Distance Metrics and Evaluation

We quantify the separation between two sets of points, denoted as $\mathcal{X}$ and $\mathcal{Y}$, through the application of conventional one-sided and double-sided $\ell_2$ Chamfer distances, denoted as $d_{\vec{C}}$, $d_C$, and Hausdorff distances, denoted as $d_{\vec{H}}$, $d_H$. The definitions for each are as follows:

$$d_{\vec{C}}(\mathcal{X}, \mathcal{Y}) = \frac{1}{|\mathcal{X}|} \sum_{\boldsymbol{x} \in \mathcal{X}} \min_{\mathbf{y} \in \mathcal{Y}} \|\boldsymbol{x} - \mathbf{y}\|_2, \qquad d_C(\mathcal{X}, \mathcal{Y}) = \frac{1}{2} \left( d_{\vec{C}}(\mathcal{X}, \mathcal{Y}) + d_{\vec{C}}(\mathcal{Y}, \mathcal{X}) \right)$$

$$d_{\vec{H}}(\mathcal{X}, \mathcal{Y}) = \max_{\boldsymbol{x} \in \mathcal{X}} \min_{\mathbf{y} \in \mathcal{Y}} \|\boldsymbol{x} - \mathbf{y}\|_2 \qquad d_H(\mathcal{X}, \mathcal{Y}) = \max \left\{ d_{\vec{H}}(\mathcal{X}, \mathcal{Y}) + d_{\vec{H}}(\mathcal{Y}, \mathcal{X}) \right\}.$$

The estimation of the distance from the INR to the target point clouds is performed identically as in the original paper (Park et al., 2023) by first creating a mesh by extracting the zero level set of $u$ using the marching cubes algorithm Lorensen & Cline (1987) on a $512 \times 512 \times 512$ uniform grid, then by sampling $10^7$ points uniformly from the surface and finally by measuring the presented distances from the sampled points and the target points cloud.

Furthermore, to measure the accuracy of the trained gradient field, normal consistency (NC) is evaluated between the learned $G$ and the surface normal from a given oriented point cloud $\mathcal{X}, \mathcal{N} = \{\boldsymbol{x}_i, \mathbf{n}_i\}_{i=1}^{N}$ comprising of sampled points $\boldsymbol{x}_i$ and the corresponding outward normal vectors $\mathbf{n}_i$, NC is defined by the average of the absolute dot product of the trained $G$ and the surface normals.

$$NC(G, \mathcal{X}, \mathcal{N}) = \frac{1}{N} \sum_{i=1}^{N} \left| G(\boldsymbol{x}_i)^{\mathrm{T}} \mathbf{n}_i \right| \tag{7}$$

### 3.4 Datasets

The SRB dataset (Berger et al., 2013) is used to benchmark this re-implementation against the original results. It consists of five different benchmark figures. Each of them has a scan point cloud and a ground truth point could. The scan is used for training the reconstruction models while the ground truth is used to evaluate the trained model.

An evaluation of the Thingy10K Zhou & Jacobson (2016) dataset was also initially planned, but as the specific objects used for experiments by Park et al. (2023) were not specified in detail, it was not possible to evaluate our implementation on this dataset without substantial effort to recover the 3D models used in the original paper.

All scanned data points are utilized as boundary points in the loss function. To enhance consistency, neighboring points around these boundary points are sampled by adding a normally distributed variable with a variance equal to the distance to their 50th closest neighbor within the scanned dataset. One local point is sampled per scanned point in the batch of scan points.

To ensure that the model has been trained on points representing the entire data distribution, 2048 points are uniformly sampled from the cubic space $[-1.1, 1.1]^3$ at each training step.

### 3.5 Hyperparameters

The numerous hyperparameters of the method are all configured to match the experimental setup of the original paper. However, one notable exception is the epsilon parameter in the loss function. In the source

code, it is suggested that using a value of $\epsilon = 0.1$ might yield better results than the original value of $\epsilon = 1$, leading us to test both values. The appendix figure 2 presents how the function $\delta_\epsilon(x) = 1 - \tanh^2\left(\frac{x}{\epsilon}\right)$ depends on $\epsilon$. The loss weight hyperparameters of the loss terms were set to $\lambda_1 = 0.1, \lambda_2 = 10^{-4}, \lambda_3 = 5 \cdot 10^{-4}$ and $\lambda_4 = 0.1, p = \infty$ and $F = \frac{x}{3}$ as in the original paper. The Adam (Kingma & Ba, 2014) optimizer is used with a learning rate of $10^{-3}$ and a learning rate schedule that reduces the learning rate by a factor of 0.99 every 2000 step. It has to be noted that this schedule is a fraction of a reduction in learning rate and probably has an insignificant impact on training ($0.99^{\frac{100000}{2000}} = 0.99^5 \approx 0.95$). The model is trained for 100 000 steps with a batch size of 16384. The network architecture consists of 7 layers with 512 hidden dimensions and skip connections from the first layer to the fourth layer. The softplus activation function is used with a beta parameter of 100.0 which makes it very close to a ReLU activation function.

## 3.6 Experimental Setup

The experiments involved training the model depicted in Figure 1 using the loss function specified in Equation 6 and hyperparameters detailed in Section 3.5. Training was conducted on the entire SRB dataset, and the trained networks were subsequently evaluated using all metrics outlined in Section 3.3. To provide robust estimates, these steps were iterated three times to gauge the uncertainty of the training process.

Implementation of the model and training loop was carried out using JAX Bradbury et al. (2018). Initially, the implementation strictly followed the instructions provided in the original paper. Subsequently, to ensure accuracy and consistency, the published code [2] was consulted for additional guidance and verification.

Consistent with the methodology employed by Park et al. (2023), all experiments were conducted on a single NVIDIA® RTX 3090 GPU with 24GB of memory.

## 3.7 Performance Optimizations

Multiple steps were taken to increase the performance of the re-implementation of the training code. In PyTorch Paszke et al. (2017), the computational graph is built dynamically and the low-level operations are dispatched on the fly during the training process, which can hamper execution speed but eliminates the need for a compilation step and enables flexibility. In the re-implementation, all the operations of the training process are expressed using JAX primitives such as jax.lax.scan, which can be compiled using XLA. The computational graph in JAX is built during the just-in-time compilation step and then transformed into low-level instructions using the XLA compiler, which results in more optimized low-level instructions and faster run times in general, sacrificing the dynamism of the PyTorch code. This re-implementation is therefore considerably faster as every operation part of the training loop has static shapes and is transformed into low-level instructions using the XLA compiler.

Furthermore, to calculate the total loss at each training step, the PyTorch code used in the original paper makes 11 different calls to the `torch.autograd.grad` function, which calculates the gradient of the outputs with respect to the inputs. On the other hand, the re-implementation combines these 11 calls to `torch.autograd.grad` into one vectorized call to `jax.jacfwd`. Forward-mode automatic differentiation with `jax.jacfwd` was chosen to replace the 11 different calls to the backward-mode automatic differentiation with `torch.autograd.grad` as the number of outputs is larger than the number of inputs.

## 3.8 Correctness of the Reproducibility

A numerical verification of the correctness of the re-implementation is conducted [3] by comparing the results of a forward and backward pass using both the re-implemented code and the code of the original paper. The weights of the neural network are initialized using the code of the original paper and then converted to JAX arrays to use in the re-implemented code. 16384 random points are then passed to both networks and the maximum difference between the outputs is $10^{-7}$. To verify that the loss is calculated correctly a difference of the gradients of the model parameters with respect to the loss of the two models is calculated using JAX

---

[2]https://github.com/yebbi/pinc
[3]https://anonymous.4open.science/r/pinc-B7CD/scripts/forward_pass_comparison.py

and PyTorch autodiff and the maximum difference is measured to less than $10^{-6}$. It is therefore reasonable to assume that the implementations are the same.

The two training implementations differ due to the calculation of the distance to the 50th nearest neighbor that in the code for the original paper mistakenly includes the normals together with the coordinates in the distance calculation, which influences the standard deviation of the boundary points for creating local filler points. This difference is detailed in the code's README [4].

## 4    Results

Table 1: Distance results for the SRB dataset using different surface reconstruction methods. The re-implementation is labeled as "[Re] PINC", the code of the original paper running on our hardware is labeled as "PINC (rerun)", and the results presented in the original paper are labeled as "PINC". The results of the other methods are sourced from their respective papers. The ground truth data is referred to as "GT", while the training data is referred to as "Scan".

| | Anchor | | | | Daratech | | | | Dc | | | | Gargoyle | | | | Lord quas | | | |
| | GT | | Scan | | GT | | Scan | | GT | | Scan | | GT | | Scan | | GT | | Scan | |
| | $d_C$ | $d_H$ | $d_{\vec{C}}$ | $d_{\vec{H}}$ | $d_C$ | $d_H$ | $d_{\vec{C}}$ | $d_{\vec{H}}$ | $d_C$ | $d_H$ | $d_{\vec{C}}$ | $d_{\vec{H}}$ | $d_C$ | $d_H$ | $d_{\vec{C}}$ | $d_{\vec{H}}$ | $d_C$ | $d_H$ | $d_{\vec{C}}$ | $d_{\vec{H}}$ |
|---|---|---|---|---|---|---|---|---|---|---|---|---|---|---|---|---|---|---|---|---|
| IGR | 0.45 | 7.45 | 0.17 | 4.55 | 4.90 | 42.15 | 0.70 | 3.68 | 0.63 | 10.35 | 0.14 | 3.44 | 0.77 | 17.46 | 0.18 | 2.04 | 0.16 | 4.22 | 0.08 | 1.14 |
| SIREN | 0.72 | 10.98 | 0.11 | 1.27 | 0.21 | 4.37 | 0.09 | 1.78 | 0.34 | 6.27 | 0.06 | 2.71 | 0.46 | 7.76 | 0.08 | 0.68 | 0.35 | 8.96 | 0.06 | **0.65** |
| SAL | 0.42 | 7.21 | 0.17 | 4.67 | 0.62 | 13.21 | 0.11 | 2.15 | 0.18 | 3.06 | 0.08 | 2.82 | 0.45 | 9.74 | 0.21 | 3.84 | 0.13 | 414.00 | 0.07 | 4.04 |
| PHASE | **0.29** | 7.43 | **0.09** | 1.49 | 0.35 | 7.24 | **0.08** | **1.21** | 0.19 | 4.65 | 0.05 | 2.78 | 0.17 | 4.79 | 0.07 | 1.58 | 0.11 | **0.71** | 0.05 | 0.74 |
| DiGS | **0.29** | 7.19 | 0.11 | **1.17** | **0.20** | **3.72** | 0.09 | 1.80 | 0.15 | **1.70** | 0.07 | 2.75 | 0.17 | **4.10** | 0.09 | 0.92 | 0.12 | 0.91 | 0.06 | 0.70 |
| PINC | **0.29** | 7.54 | **0.09** | 1.20 | 0.37 | 7.24 | 0.11 | 1.88 | **0.14** | 2.56 | **0.04** | 2.73 | **0.16** | 4.78 | **0.05** | **0.80** | **0.10** | 0.92 | **0.04** | 0.67 |
| PINC (rerun) $\varepsilon = 1$ | 0.32 | 7.63 | 0.10 | 1.26 | 5.97 | 55.25 | 0.64 | 4.38 | 0.15 | 2.62 | 0.05 | 2.80 | **0.16** | 4.77 | 0.06 | 0.81 | 0.11 | 0.79 | **0.04** | 0.74 |
| PINC (rerun) $\varepsilon = 0.1$ | 0.37 | 9.10 | 0.10 | 3.94 | 7.48 | 62.82 | 0.66 | 7.39 | 0.16 | 2.26 | 0.06 | 2.75 | 0.19 | 5.95 | 0.07 | 3.86 | 0.14 | 3.41 | 0.05 | 1.50 |
| [Re] PINC $\varepsilon = 1$ | 0.31 | **6.01** | 0.17 | 1.41 | 4.73 | 53.51 | 0.48 | 3.42 | 0.17 | 2.18 | 0.10 | 2.73 | 0.21 | 4.67 | 0.14 | 0.91 | 0.18 | 1.81 | 0.12 | 0.86 |
| [Re] PINC $\varepsilon = 0.1$ | 0.35 | 7.67 | 0.16 | 1.39 | 7.52 | 69.12 | 0.24 | 3.10 | 0.17 | 2.37 | 0.11 | **2.72** | 0.21 | 5.25 | 0.13 | 1.36 | 0.17 | 1.17 | 0.12 | 1.01 |

The published results of various comparable surface reconstruction methods applied to the SRB dataset Gropp et al. (2020); Sitzmann et al. (2020); Atzmon & Lipman (2020); Lipman (2021); Ben-Shabat et al. (2022); Park et al. (2023) compared to the results obtained in this study are presented in Table 1. Notably, our replication of the model exhibits superior performance compared to most other models for all benchmarks except Daratech which fails to converge to a closed surface. However, it falls short of achieving the performance levels reported in the original paper. Interestingly, upon training the model using the code from the original paper, this study obtains results comparable to those of the re-implemented models. A significant observation is the consistently inadequate performance of both evaluated models in the case of Daratech, where no convergence to a stable solution could be measured. This can be visually confirmed in 6 and the observation holds for both this re-implementation and the models trained with the code used in the original paper.

Furthermore, the training with a smaller $\varepsilon$ produces a less smooth mesh which means that $d_{\vec{C}}$ improves while $d_H$ deteriorates.

The normal consistency 7 of the trained models is compared to the normal consistency of the other methods reported by Park et al. (2023) in Table 2. It is apparent that the evaluation results of the re-implementation are higher than those of similar methods except for the Daratech reconstruction. It is not entirely clear why this is the case and since Park et al. (2023) did not publish their evaluation code or their trained model, it is not possible to verify that the evaluation implementation is numerically identical.

All reconstructed surfaces are evaluated using sampling of a mesh constructed with marching cubes, which means that the evaluation metrics vary for each calculation. Furthermore, the evaluation code is not included in the code for the original paper, making the utilization of an identical random seed impossible. The variation due to the surface sampling during the evaluation over 15 different evaluation runs is presented in Table 3, which shows that this variation is negligible.

---

[4] https://anonymous.4open.science/r/pinc-B7CD/README.md

Table 2: Normal consistency of reconstructed surfaces on SRB. The values for other surface reconstruction are those reported by Park et al. (2023).

|  | Anchor | Daratech | Dc | Gargoyle | Lord quas |
|---|---|---|---|---|---|
| IGR | 0.9706 | 0.8526 | 0.9800 | 0.9765 | 0.9901 |
| SIREN | 0.9438 | **0.9682** | 0.9735 | 0.9392 | 0.9762 |
| DiGS | 0.9767 | 0.9680 | 0.9826 | 0.9788 | 0.9907 |
| SAP | 0.9750 | 0.9414 | 0.9636 | 0.9731 | 0.9838 |
| PINC | 0.9754 | 0.9311 | 0.9828 | 0.9803 | 0.9915 |
| [Re] PINC $\varepsilon = 1$ | 0.9864 | 0.8595 | **0.9940** | 0.9915 | 0.9962 |
| [Re] PINC $\varepsilon = 0.1$ | **0.9874** | 0.8722 | **0.9940** | **0.9917** | **0.9963** |

Table 3: Uncertainty estimation from the random sampling during the evaluation step. Mean and standard deviation over 15 different evaluation runs using different random seeds and $10^7$ surface sample points, as in Park et al. (2023). The evaluated runs are using the reimplemented JAX code.

| | | GT | | Scan | |
|---|---|---|---|---|---|
| | | $d_C$ | $d_H$ | $d_{\overrightarrow{C}}$ | $d_{\overrightarrow{H}}$ |
| Anchor | $\varepsilon = 1$ | $0.31157 \pm 0.00005$ | $6.01387 \pm 0.00012$ | $0.17076 \pm 0.00001$ | $1.41510 \pm 0.00024$ |
| | $\varepsilon = 0.1$ | $0.34599 \pm 0.00007$ | $7.67212 \pm 0.00002$ | $0.16095 \pm 0.00001$ | $1.38895 \pm 0.00011$ |
| Daratech | $\varepsilon = 1$ | $4.72754 \pm 0.00157$ | $53.49058 \pm 0.01563$ | $0.47939 \pm 0.00001$ | $3.41532 \pm 0.00010$ |
| | $\varepsilon = 0.1$ | $7.52066 \pm 0.00249$ | $69.10534 \pm 0.01205$ | $0.24270 \pm 0.00002$ | $3.10309 \pm 0.00160$ |
| Dc | $\varepsilon = 1$ | $0.16654 \pm 0.00024$ | $2.18049 \pm 0.00052$ | $0.09928 \pm 0.00018$ | $2.73660 \pm 0.00175$ |
| | $\varepsilon = 0.1$ | $0.17487 \pm 0.00002$ | $2.37276 \pm 0.00006$ | $0.10850 \pm 0.00002$ | $2.72449 \pm 0.00011$ |
| Gargoyle | $\varepsilon = 1$ | $0.20605 \pm 0.00003$ | $4.66616 \pm 0.00019$ | $0.13511 \pm 0.00001$ | $0.90959 \pm 0.00098$ |
| | $\varepsilon = 0.1$ | $0.20768 \pm 0.00002$ | $5.25331 \pm 0.00010$ | $0.12680 \pm 0.00001$ | $1.36158 \pm 0.00036$ |
| Lord quas | $\varepsilon = 1$ | $0.18022 \pm 0.00002$ | $1.80669 \pm 0.00006$ | $0.12490 \pm 0.00001$ | $0.85823 \pm 0.00023$ |
| | $\varepsilon = 0.1$ | $0.17083 \pm 0.00001$ | $1.17148 \pm 0.00009$ | $0.12495 \pm 0.00001$ | $1.01274 \pm 0.00010$ |

Table 4 presents the mean values and 95% confidence intervals of the metrics, calculated with the Student's t-test, for three training runs with different random seeds. The discrepancies between the originally reported results in Table 1 and the re-implementation results with confidence intervals in Table 4 suggest significant differences, raising questions about potential methodological errors or the influence of selective reporting in the original paper. However, the confidence intervals of our findings indicate that these variations are unlikely to be attributed to mere chance or "bad luck." Furthermore, none of the previous papers did not include results with confidence bounds using multiple training runs.

The training times of the code used in the original paper and this re-implementation are examined in Table 5. It can be noted that the re-implementation requires approximately 5 less training time compared to the code used in the original paper.

A visualization of all the final reconstructed surfaces is presented in the appendix figure 6, highlighting the failure of converging for the Daratech example by the trained models, all with the hyperparameters reported by Park et al. (2023).

## 5 Reproducibility Discussion

Replicating various aspects of the paper presented both straightforward and challenging elements.

Firstly, the paper provided detailed explanations of the $p$-Poisson equation and the incorporation of the curl-free constraint, which simplified the implementation process of the equations.

Furthermore, the experimental setup section offered details about the neural network architecture, the variable-splitting strategy, the use of auxiliary variables, the proposed loss function, the integration of the minimal area criterion, and the explicit handling of occlusions, which facilitated the rewriting of the model and forward pass using JAX primitives.

The use of Chamfer and Hausdorff distances as evaluation metrics made it possible to benchmark the reproduced results against the original findings and to compare them to the metrics reported by other

Table 4: Uncertainty estimation of the distance metrics and normal consistency from three training runs compared to the results reported in the original paper. Mean and t-student confidence interval of 95% over three different training runs using different random seeds are reported. The rows labeled 'PINC' present the reported metrics in the original paper and the rows with confidence intervals present results from three training runs using the reimplemented JAX code with two different values of $\varepsilon$.

| | | GT | | Scan | | |
|---|---|---|---|---|---|---|
| | | $d_C$ | $d_H$ | $d_{\overrightarrow{C}}$ | $d_{\overrightarrow{H}}$ | NC |
| Anchor | PINC | 0.290 | 7.540 | 0.090 | 1.200 | 0.975 |
| | $\varepsilon=1$ | $0.315 \pm 0.007$ | $6.046 \pm 0.073$ | $0.167 \pm 0.008$ | $1.404 \pm 0.041$ | $0.987 \pm 0.000$ |
| | $\varepsilon=0.1$ | $0.350 \pm 0.010$ | $7.787 \pm 0.564$ | $0.163 \pm 0.011$ | $1.399 \pm 0.070$ | $0.987 \pm 0.000$ |
| Daratech | PINC | 0.370 | 7.240 | 0.110 | 1.880 | 0.931 |
| | $\varepsilon=1$ | $3.782 \pm 2.124$ | $45.591 \pm 17.924$ | $0.497 \pm 0.152$ | $3.491 \pm 0.449$ | $0.854 \pm 0.012$ |
| | $\varepsilon=0.1$ | $5.876 \pm 10.483$ | $50.889 \pm 85.656$ | $0.391 \pm 0.334$ | $3.392 \pm 0.735$ | $0.873 \pm 0.007$ |
| Dc | PINC | 0.140 | 2.560 | 0.040 | 2.730 | 0.983 |
| | $\varepsilon=1$ | $0.173 \pm 0.013$ | $2.134 \pm 0.473$ | $0.106 \pm 0.015$ | $2.722 \pm 0.042$ | $0.994 \pm 0.001$ |
| | $\varepsilon=0.1$ | $0.176 \pm 0.003$ | $2.358 \pm 0.366$ | $0.109 \pm 0.001$ | $2.722 \pm 0.006$ | $0.993 \pm 0.002$ |
| Gargoyle | PINC | 0.160 | 4.780 | 0.050 | 0.800 | 0.980 |
| | $\varepsilon=1$ | $0.203 \pm 0.007$ | $4.567 \pm 0.235$ | $0.134 \pm 0.004$ | $0.968 \pm 0.134$ | $0.991 \pm 0.001$ |
| | $\varepsilon=0.1$ | $0.209 \pm 0.006$ | $5.224 \pm 0.068$ | $0.132 \pm 0.014$ | $1.342 \pm 0.279$ | $0.992 \pm 0.000$ |
| Lord quas | PINC | 0.100 | 0.920 | 0.040 | 0.670 | 0.992 |
| | $\varepsilon=1$ | $0.175 \pm 0.016$ | $1.514 \pm 1.173$ | $0.125 \pm 0.001$ | $0.871 \pm 0.029$ | $0.996 \pm 0.000$ |
| | $\varepsilon=0.1$ | $0.170 \pm 0.010$ | $1.179 \pm 0.056$ | $0.125 \pm 0.008$ | $0.958 \pm 0.177$ | $0.996 \pm 0.000$ |

Table 5: Comparison of total training times for 100 000 steps.

| | Total Training Time (hours) |
|---|---|
| Code used in the original paper | 13.5 |
| Re-implemented code | 2.2 |

comparable implicit neural representation methods for surface reconstruction. The reliance on the widely used SRB dataset (Berger et al., 2013) enabled a direct comparison of the performance of the reproduced model with the results reported in the original paper, without the need for additional data pre-processing or acquisition.

However, several challenges were encountered during the reproduction process.

Multiple undocumented numerical details were not mentioned in the original paper. The number of sampled global points is not mentioned in the paper and there is an undocumented division by $\sqrt{2}$ after the skip connection in the multi-layer perceptron. Furthermore, the geometric initialization is also slightly different between implementations. In IGR Gropp et al. (2020), the weights are initialized with $\mathcal{N}(\sqrt{\pi/n_{dim}}, 10^{-6})$ and the bias is initialized to $-1$ [5], in SAL Atzmon & Lipman (2020), the weights are the weights are initialized with $\mathcal{N}(2\sqrt{\pi/n_{dim}}, 10^{-6}))$ and the bias is initialized to $-1$ [6], and in PINC Park et al. (2023), the weights are initialized with $\mathcal{N}(\sqrt{\pi/n_{dim}}, 10^{-6}))$ and the bias is initialized to $-0.1$ [7].

From the results obtained in this study, it is apparent that training models using the code of the original paper result in distance values comparable to those of this re-implementation. However, these distance values are not on par with those presented in Park et al. (2023). As the confidence intervals of our findings indicate that these variations are unlikely to be attributed to mere chance or "bad luck", this raises questions about potential methodological errors or the influence of selective reporting in the original paper.

Another encountered challenge, was that code used for the original paper has very long training times, which means that substantial computational resources are needed to run the published code for comparison.

An e-mail was sent to the authors (Park et al., 2023) inquiring about the availability of evaluation code, trained models, and reconstruction results but has not been answered at the time of writing.

---

[5] https://github.com/amosgropp/IGR/blob/master/code/model/network.py#L48
[6] https://github.com/matanatz/SAL/blob/master/code/model/network.py#L112
[7] https://github.com/Yebbi/PINC/blob/main/model/network.py#L46

# 6    Conclusion

In this study, the primary objective was to replicate and verify the findings of the implicit neural representation surface reconstruction method proposed by Park et al. (2023). The original study introduces a novel technique that integrates the $p$-Poisson equation into the model's loss function, enabling surface reconstruction from point clouds without relying on additional information such as surface normals. This purportedly leads to substantial improvements over comparable implicit neural representations.

The results of this study reveal that while the surface reconstruction method delineated in the original paper surpasses numerous alternative approaches, it significantly fails to achieve the distance metrics initially reported when subjected to rerunning or re-implementation as part of this study. Notably, when utilizing the code from the original paper for training, we observed discrepancies in performance compared to the reported distance metrics. The numerical comparisons of the model, loss, and gradients from the original paper and this re-implementation demonstrate identical numerical values up to $10^{-6}$.

Finally, the re-implementation of the method surpassed the training speed of the code used in the original paper, resulting in a remarkable five-fold speedup in training times. This reveals that the adoption of just-in-time compilers such as the one in JAX for implementing machine learning algorithms and optimizing computational tasks can considerably increase computational efficiency and the speed of training. This gives an actionable insight for others to explore and integrate such compilers into their machine-learning workflow, with a potential for significant improvements in research output, especially for resource-intensive computational tasks.

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

# A   Appendix

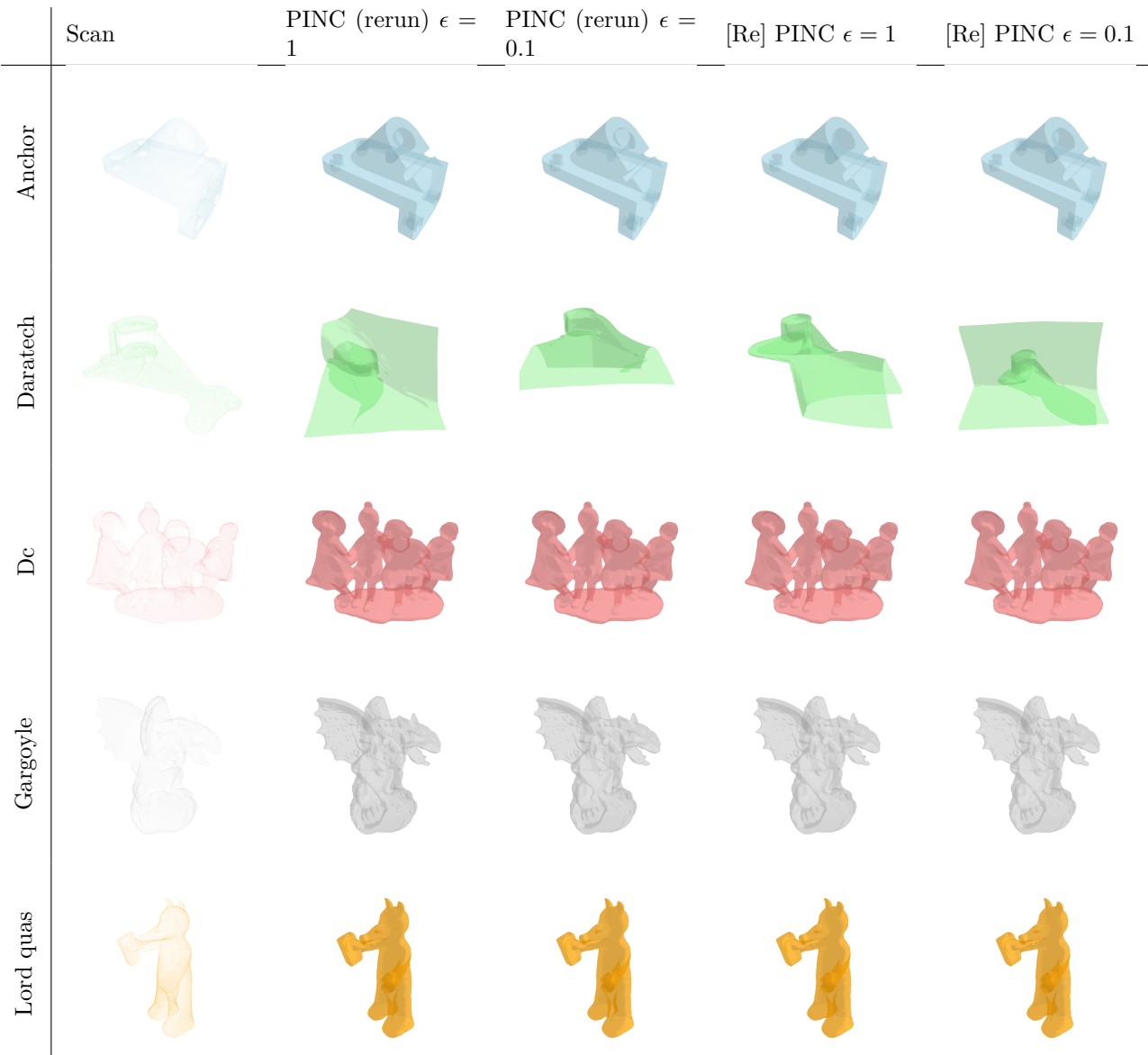

Table 6: 3D Reconstruction results for SRB Dataset.

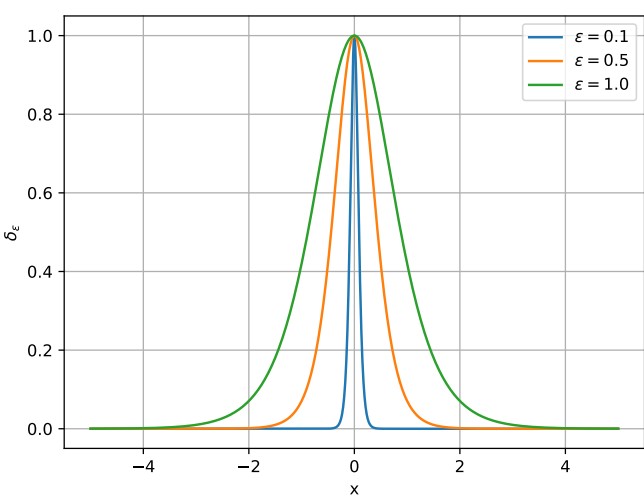

Figure 2: Plot of the function $\delta_\epsilon(x) = 1 - \tanh^2\left(\frac{x}{\epsilon}\right)$ for different values of $\epsilon$

