# OpenReview forum: "[Re] $p$-Poisson surface reconstruction in curl-free flow from point clouds"
_TMLR — Rejected by TMLR_

### Review · Reviewer_anWF · 2024-03-17

**Summary Of Contributions:**

The paper presents a reproducibility study of "p-Poisson surface reconstruction in curl-free flow from point clouds", focusing on validating the original claims. The authors provide an efficient reimplementation of the original paper that is 5x faster and conclude that the trends of the original paper are generally reproducible. The numerical results do not agree exactly with the original paper, which is explained by possibly different model weights, the lack of evaluation code, and possibly different hyperparameters due to the lack of some experimental details in the original paper. Furthermore, the authors seem to try to get details from the authors of the original paper, but could not reach them.

**Audience:**

Yes

**Broader Impact Concerns:**

This work does not contain any broader impact concern.

**Claims And Evidence:**

Yes

**Requested Changes:**

Please refer to the weakness above.

**Strengths And Weaknesses:**

**Strengths**

* The paper systematically reviews the reproducibility of the original paper. Scope is clearly stated. The authors appear to have made an effort to communicate with the authors of the original paper.

* The authors tackle one of the primary drawbacks of the original paper: lengthy training times and the utilization of resource-intensive computation. Their reimplementation leads to a noteworthy enhancement in efficiency, with 5x faster training times, which could be valuable for future research in this area.

**Weaknesses**
*  The main contribution of the paper seems to the reimplementation of PINC, which drives up the computational efficiency, but they didn't discuss why. The speedup appears to be due to efficient numerical computations and automatic differentiation using JAX, but it is unclear whether this improvement is specific to this paper or applicable to implicit neural representation models using auto diff in general.

*  The quantitative results are different from the original paper, but this is most likely due to unavailability of complete original experimental setups and evaluation code. I appreciate the effort of the authors to compare the metrics in Table 1, but in surface reconstruction, Hausdorff/Chamfer distances do not reflect all the quality of the surface. Therefore, it is also important to look at it qualitatively. But, the quality of the figures in Table 3 of the manuscript are not good enough to judge the results (especially, all the details are lost).

*  The essence of PINC lies in the use of the p-Poisson equation and variable splitting to resolve the ill-posedness of the eikonal equation and to eliminate the model dependence on the surface normal vector. While the authors evaluate the reproducibility of the original model on the SRB dataset and show that it leads to decent reconstructions without surface normals, I think the experiments are insufficient to validate the main claims of PINC. For example, the authors list normal consistency in Section 4.3 as a metric to evaluate the strength/effect of variable splitting, but do not actually measure it.

*  I am concerned about their limited novelty and contributions, as they merely replicate the original paper without any additional experiments or rectification of the weaknesses of the original model.

---

### Review · Reviewer_ddPG · 2024-03-31

**Summary Of Contributions:**

This paper provides a reproducibility analysis of the p-Poisson surface reconstruction method presented by Park et al. The authors used JAX to reimplement the method and achieved faster training performance. The reimplemented results are very close to the offcial code. However, this paper points out that the results still deviate from the original reported numbers.

**Audience:**

Yes

**Claims And Evidence:**

Yes

**Requested Changes:**

In addition to the methodology, I recommend mentioning crucial implementation details and conducting a side-by-side analysis of them. For example, if an implementation in the official code is not the same as described or is unclear in the original paper? How would you address that?

I encourage the authors to run the experiments in table 1 multiple times and demonstrate the means and variations of the results.

**Strengths And Weaknesses:**

The authors have developed a reimplemented version of this method, investing significant engineering efforts. This version is efficient and achieves a fivefold increase in training speed compared to the official code. I think it will be beneficial for future research.

Meanwhile, this paper has some weaknesses in the description and experiments. Although the authors have described the overall methodology, many implementation details that may affect the results are not clearly specified in the paper. In the comparisons, the authors only use a single run to report the numbers in table 1. It is not very clear whether the training initialization will have a significant impact.

---

### Review · Reviewer_8XW4 · 2024-04-14

**Summary Of Contributions:**

This is a reproducibility study, aiming to replicate a work presented at NeurIPS 2023 that originally developed a method for neural-implicit surface reconstruction from unstuctured point clouds, based on a variant of Poisson surface reconstruction. There is some discussion of issues encountered during the reimplementation. The presented results include those from the original paper, those from the code released by the authors of the original paper, and those from a reimplementation by the authors of this study. It is shown that the results in the original paper are not equal to those from either implementation.

**Audience:**

No

**Broader Impact Concerns:**

None -- this work does not raise any significant concerns (nor is there any broader impact statement present)

**Claims And Evidence:**

Yes

**Requested Changes:**

- Analyse and discuss the stochastic variation of both original and new implementations; specify the error bars / standard deviations on all results, so it becomes clear whether (i) the non-reproducibility is statistically significant; (ii) whether the original claims were statistically significant. This includes stochasticity both during training and evaluation.
- Clarify how the "identical numerical results" for a forward pass degenerate into significantly different quantitative results for the overall task
- Rework the method recap to be clearer, explaining why the p-Poisson equation was used, why G was introduced (as opposed to using autodiff on u), etc.
- Consider merging sec. 1 with the subsequent two sections; in particular, to have this "extended abstract" followed by a still very short introduction & scope seems unnecessary.
- Typo: mid p4: "How to neural network structure" → "How the…"

**Strengths And Weaknesses:**

Strengths
- The scope of the reproducibility study is clearly described
- There is helpful discussion on what aspects were easy vs difficult to implement
- The study provides some new information, i.e. that the original implementation of the method does not yield results matching the original paper, and mentions certain discrepancies between the paper and code that might otherwise be overlooked
- The authors release a Jax re-implementation that is approximately six times faster than the Pytorch original
- Overall the paper is clear and readable throughout

Weaknesses
- The study yields little insight, beyond the mere fact that the original implementation's results do not match the original paper. In particular, the dataset used is the same as the original (in fact only one of the two in the original study), and there are no "actionable lessons" (per the TMLR acceptance criteria). There is no attempt to explore interesting questions such as degrees of robustness to different kinds/magnitudes of noise in the point-clouds, robustness to missing regions, etc. If the original implementation were unavailable, I would consider this somewhat less problematic, but since the original implementation is available and (out-of-the-box) yields results comparable with this reimplementation, it is difficult to see what new scientific knowledge is added by this study.
- There is no consideration of statistical significance – i.e. whether the results from the reimplementation are significantly different from those from the original – and same for original code vs paper, and indeed everything vs baselines.
- It is unclear if there is any stochasticity in the replicated or original implementation, and whether identical random numbers were used if so (this would be challenging given the use of different frameworks – pytorch & jax). This makes it impossible to judge whether the non-replicating results are simply due to bad luck (and maybe the original authors cherry-picking the best from many runs), or a genuine methodological issue.
- It is claimed that the reimplementation yields "identical numerical results" to the original code, yet the quantitative results are somewhat different (albeit considerably less different than the results from the original paper). There is no discussion or analysis of this important discrepancy. Presumably either the gradients differ (was this checked? only forward is mentioned), or the optimisation diverges at some point from the original (when if so?), maybe due to cuda nondeterminism (was this controlled for?).
- The structure of the method recap is a bit tangled; in particular it is only mentioned that G is supposed to approximate the gradient when discussing the zero-curl condition, not when introducing G; also it is not mentioned that a variable-splitting strategy is being used.

---

### Decision · Action_Editor_JMyX · 2024-05-22

**Recommendation:** Reject

**Comment:**

Altogether, the reviewers appreciate some discussions provided by the authors, the new implementation, with the corresponding speedup. They feel, however, that this rather corresponds to an engineering effort than a scientific contribution.

More importantly, and as stated above, the reviewers all express concerns regarding the limited insights provided by the authors to explain the differences between their results and those reported by the authors of the original paper. The reviewers also argue that these differences may not be statistically significant.

As a result, all reviewers recommended rejection, and the AE concurs with them that this submission does not meet the TMLR acceptance threshold.

**Audience:**

No. All three reviewers argue, both in their initial reviews and in their final recommendation, that the manuscript does not provide sufficient insight as to why the authors observed discrepancies between their results and those of the authors from the original paper. Ultimately, and as stated by Reviewer 8XW4, the paper lacks "actionable lessons" as per the TMLR acceptance criteria.

**Claims And Evidence:**

No. In particular, the reviewers expressed concerns regarding the statistical significance of the results. The standard deviations reported in the revised version of the manuscript failed to convince the reviewers, as explicitly mentioned by Reviewer 8XW4 in their final recommendation.